# Early Changes of the Standardized Uptake Values (SUV_max_) Predict the Efficacy of Everolimus-Exemestane in Patients with Hormone Receptor-Positive Metastatic Breast Cancer

**DOI:** 10.3390/cancers12113314

**Published:** 2020-11-10

**Authors:** Marianna Sirico, Ottavia Bernocchi, Navid Sobhani, Fabiola Giudici, Silvia P. Corona, Claudio Vernieri, Federico Nichetti, Maria Rosa Cappelletti, Manuela Milani, Carla Strina, Valeria Cervoni, Giuseppina Barbieri, Nicoletta Ziglioli, Martina Dester, Giulia Valeria Bianchi, Filippo De Braud, Daniele Generali

**Affiliations:** 1Department of Surgery and Cancer, Imperial College London, London W12 0NN, UK; sil.corona@hotmail.it (S.P.C.); mr.cappelletti@asst-cremona.it (M.R.C.); m.milani@asst-cremona.it (M.M.); c.strina@asst-cremona.it (C.S.); c.valeria@asst-cremona.it (V.C.); g.barbieri@asst-cremona.it (G.B.); n.ziglioli@asst-cremona.it (N.Z.); m.dester@ospedale.cremona.it (M.D.); d.generali@units.it (D.G.); 2Azienda Socio-Sanitaria Territoriale Cremona, 26100 Cremona, Italy; 3Department of Medical, Surgery and Health Sciences, University of Trieste, 34147 Trieste, Italy; obernocchi@units.it (O.B.); fgiudici@units.it (F.G.); 4Section of Epidemiology and Population Science, Department of Medicine, Baylor College of Medicine, Baylor Plaza, Houston, TX 77030, USA; 5Department of Medical Oncology, Fondazione IRCCS Istituto Nazionale dei Tumori, 20133 Milan, Italy; claudio.vernieri@istitutotumori.mi.it (C.V.); federico.nichetti@istitutotumori.mi.it (F.N.); giulia.bianchi@istitutotumori.mi.it (G.V.B.); filippo.debraud@istitutotumori.mi.it (F.D.B.); 6Fondazione Istituto FIRC di Oncologia Molecolare (IFOM), 20129 Milan, Italy; 7Department of Oncology and Hemato-Oncology, University of Milan, 20122 Milan, Italy

**Keywords:** metastatic breast cancer, everolimus, ^18^F-FDG PET/CT, ∆SUV%, predictive biomarker

## Abstract

**Simple Summary:**

The combination of everolimus and exemestane was FDA approved after BOLERO-2 clinical trial results for the treatment of hormone receptor-positive (HR+) human epidermal growth factor receptor 2-negative (HER2−) metastatic breast cancer (HR+ mBC), progressing on a prior therapy with a non-steroideal aromatase inhibitor. However, there are no predictive biomarkers for tumor sensitivity or resistance to everolimus-based treatment. The aim of our retrospective study was to investigate the potential role of FDG-PET SUV (ΔSUV%) as a predictive biomarker for a long-term clinical benefit. We found in a homogenous population of 31 patients two precocious ∆SUV% thresholds capable of identifying HR+ HER2-mBC patients achieving long-term benefit or long-term survival (36 month-OS) during everolimus-exemestane therapy. Based on these results, ∆SUV, as PET-based biomarker, provides additional information on which patients are most likely to benefit from everolimus with exemestane-based therapy over a long-term period. FDG-PET is a useful and minimally invasive tool that could be used for making a decision on personal treatment enhancing benefit while reducing collateral effects.

**Abstract:**

*Background:* The mTORC1 inhibitor everolimus has been approved in combination with the aromatase inhibitor exemestane for the treatment of hormone receptor-positive (HR+) human epidermal growth factor receptor 2-negative (HER2−) metastatic breast cancer (HR+ mBC) progressing on prior therapy with a non-steroidal aromatase inhibitor. To date, no predictive biomarkers of tumor sensitivity/resistance for everolimus-based treatments have been identified. We hypothesized that precocious changes in the Standardized Uptake Volume (∆SUV%), as assessed by ^18^F-Fluorodeoxyglucosepositron-emission tomography (^18^F-FDG PET/CT), may be a marker of everolimus efficacy. *Methods:* This was a retrospective study including 31 HR+ HER2- patients treated with everolimus and exemestane in two Italian centers between 2013 and 2018. The objective of the study was to investigate ∆SUV% as a predictive marker of everolimus antitumor efficacy. ^18^F-FDG PET/CT scans were performed at baseline and after three months of treatment. Patients were defined as long responders (LRs) if disease progression occurred at least 10 months after treatment initiation and long survivors (LSs) if death occurred later than 36 months after starting therapy. ROC analysis was used to determine the optimal cut-off values of ∆SUV% to distinguish LRs from non-LRs and LSs from non-LSs. Progression-free survival (PFS) and overall survival (OS) were estimated by Kaplan–Meier method. *Results:* The SUVmax values decreased significantly from baseline to 3 months after therapy (*p* = 0.003). Dynamic changes of SUVmax (Delta SUV) had a higher accuracy in discriminating long-responders from non-long-responders (AUC = 0.67, Delta SUV cut-off = 28.8%) respects to its ability to identify long survivors from no-long survivors (AUC = 0.60, Delta SUV cut-off = 53.8%). Patients were divided into groups according to the Delta SUV cut-offs and survival outcomes were evaluated: patients with a decrease of ∆SUV% ≥ 28.8% had significantly better PFS (10 months-PFS: 63.2%, 95% CI: 37.9–80.4% and 16.7%, 95% CI: 2.7–41.3% respectively, *p* = 0.005). As regard as OS, patients with ∆SUV% ≥ 53.8% had longer OS when compared to patients with ∆SUV% < 53.8% (36 month-OS: 82.5% vs. 45.9% vs. *p* = 0.048). *Conclusion:* We found two precocious ∆SUV% thresholds capable of identifying HR+ HER2-mBC patients, which would achieve long-term benefit or long-term survival during everolimus-exemestane therapy. These results warrant further validation in prospective studies and should be integrated with molecular biomarkers related to tumor metabolism and mTORC1 signaling.

## 1. Introduction

Endocrine therapy (ET) is the mainstay of treatment of patients with hormone receptor-positive (HR+), human epidermal growth factor receptor 2-negative (HER2−) metastatic breast cancer (mBC) [1]. However, primary endocrine resistance is present in approximately 25% of HR+ HER2− mBC patients, while acquired resistance eventually occurs after a variable number of months/years after treatment initiation [2]. Therefore, endocrine-resistant HR+ HER2- mBC is still a challenge for clinicians. The phosphatidylinositol 3-kinase (PI3K)/protein kinase B (AKT)/mechanistic target of rapamycin complex 1 (mTORC1) pathway is crucially implicated in the development of acquired resistance to ET [3]. Aberrant signaling through this axis occurs in up to 70% of HR+ HER2-mBC [4]. Up-regulation of the PI3K/AKT/mTOR pathway could result from three main modalities: PI3K, AKT, and mTORC1 activating mutations, overexpression or loss of the phosphatase and tensin homolog, which is deleted from chromosome 10 (PTEN) [5]. PI3K mutations occur mainly on exons 9 (mainly c.1624G > A and c.1633G >A) and 20 (mainly c.3140 A) [6,7]. On the other hand, AKT hot spot mutation in the homologous structural domain c.49 G > A, occurs in 3% of BS and it provides greater kinase binding affinity to phosphorylated lipids of phosphatidylinositol [8,9]. PTEN protein instability and gene silencing occur in 4% to 63% of breast cancer cases [10]. PI3K mutations are the most common in BC [11].

This evidence gives the rationale for combining PI3K or mTORC1 inhibitors with different types of ETs in this setting [12]. Currently, PI3K and mTOR inhibitors as well as inhibitors of AKT (e.g., AZD5363 or GDC-0068) are in late phase clinical trials [13] (NCT01485861, NCT03012477). The phase III, randomized, BOLERO-2 trial showed that the addition of everolimus to exemestane significantly prolonged progression-free survival (PFS) in postmenopausal HR+HER2-mBC patients previously treated with one line of non-steroidal aromatase inhibitor (NSAIs) [14]. However, only a minority of patients benefited from this combination regimen, which is associated with severe adverse events in approximately 27% of patients [15].

Therefore, a clinical priority in this field is to discover non-invasive and reliable biomarkers capable of identifying patients more likely to achieve long-term benefit from everolimus-exemestane therapy, avoiding prolonged treatment and toxicities to patients who are less likely to benefit from this treatment. To date, no reliable biomarkers predictive of long-term clinical benefit from everolimus-exemestane have been identified.

The PI3K/AKT/mTORC1 axis is involved in the regulation of metabolic homeostasis and metabolic reprogramming in cancer cells, including aberrant activation of glycolysis and dysregulated biosynthetic processes. However, mTORC1 or PI3K inhibition results in systemic metabolic alterations, including hyperglycemia and hyperinsulinemia, with a range of incident toxicities between 1% and 50% [16]. These metabolic changes could affect cancer cell growth and proliferation by increasing the provision of glucose to cancer cells, and at the same time by stimulating the insulin receptor (IR)/PI3K/AKT/mTORC1 pathway and promoting tumor cell resistance to everolimus.

In the last decade, ^18^FDG Positron Emission Tomography (^18^FDG-PET) have demonstrated to be useful in breast cancer patients, specifically for the assessment of tumor response to therapy both in neoadjuvant [17] and metastatic settings [18]. Additionally, ^18^FDG-PET has been considered superior to conventional imaging procedures to define the impact of treatment in diffuse metastatic disease, especially in the skeleton and lymph node [19]. Several studies using ^18^FDG-PET showed a correlation with tumor aggressiveness but also with patients’ outcome [20]. However, it has not still been clarified whether tumor metabolic response, evaluated by FDG-PET, could be related to outcomes, in terms of overall survival (OS) and progression-free survival.

Regarding breast cancer patients treated with everolimus, only one study assessed the role of FDG-PET in predicting outcome and toxicity in mBC patients treatment with everolimus, suggesting that poor decrease in metabolic activity after 14 days of treatment can identify a subgroup of patients with a shorter PFS compared to the subgroup with a significantly lower metabolic activity [21].

Based on these reports, this study aimed at evaluating the potential role of precocious (i.e., 3 months) reduction of FDG-PET SUV (ΔSUV%) as a biomarker of long-term clinical benefit from everolimus-exemestane combination in the treatment of patients with HR+ HER2− mBC.

## 2. Results

### 2.1. Patient Characteristics

Demographic, clinical and pathological characteristics of 31 patients included in this study are presented in Table 1. The median follow-up was 17.92 months (inter quartile range: 12.96; 21.69 months). Median SUVmax of the chosen target lesion at baseline and at 3 months was 6.0 (range: 2.0; 17.2) and 3.4 (range: 0.0; 14.0), respectively (Wilcoxon paired test, *p* = 0.003) (Figure 1). Median ΔSUV (%) was +39.2% (−375.0%; +100%).

### 2.2. Determination of ΔSUV Cut-off Value to Discriminate LRs From Non-LRs and LSs From Non-LSs

ROC analysis was performed to determine the optimal cut-off value of ΔSUV (%) capable of discriminating LRs from non-LRs. The ROC curve is shown in Figure 2A. A cut-off ΔSUV (%) of 28.7% was found as the optimal value that distinguished LRs from non-LRs, with an area under the ROC curve (AUC) of 0.67 and 95% CIs of 0.46–0.89. In particular, 63.2% (95%CI: 37.9–80.4%) of LRs were progression-free at 10 months after treatment initiation, while only 16.7% of non-LRs had not undergone disease progression at the same time point (*p* = 0.005). Regarding OS, ROC analysis identified 53.8% as the optimal cut-off point for ΔSUV%, with AUC= 0.60, 95% CI: 0.39–0.81 (Figure 2B). OS at 36 months was 82.5%, (95%CI: 46.1–95.3%) in LSs and 45.9% (95% CI: 22.8–66.4%) in non-LS (*p* = 0.048, Log-Rank test). We moreover investigated the role of basal SUV as a potential indicator for later response or longer survival (36 month-OS), finding that basal SUV had a poor ability to discriminate both LRs from nLRs (AUC = 0.50 (0.29–0.72)) than LSs and LSs (AUC = 0.48 (0.26–0.69)).

### 2.3. Impact of ΔSUV (%) on PFS and OS

Based on previous analyses, patients were defined as LRs if ΔSUV (%) was higher than 28.8% (i.e., higher than 28.8% reduction of ΔSUV (%)), and as non-LRs if ΔSUV (%) was equal to or lower than 28.8% at 3-month PET scan evaluation. Using this cut-off, we identified 19(61.3%) LRs and 12(38.7%) non-LRs. Similarly, patients were defined as LSs if ΔSUV (%) was higher than 53.8% (i.e., higher than 53.8% reduction of ΔSUV (%)), and as non-LSs if ΔSUV (%) was equal to or lower than 53.8% at 3-month PET scan evaluation. Median PFS was not reached among LRs, while it was 6.3 months (95%CI: 2.6–9.4) in non-LRs (see Figure 3A). Regarding OS, median OS was not reached among LSs, while it was 27.83 months (95% CI: 12.363-NA) in non-LSs (see Figure 3B). Figure 4 showed 4 FDG-PET at baseline, after one month and after three months.

## 3. Discussion

The goal of this study was to investigate the relationship between SUV changes after 3 months of treatment and long-term efficacy of everolimus plus exemestane in patients with HR+ HER2- mBC.

We found that, as early as 3 months after treatment initiation, FDG-PET scan evaluation of FDG uptake identified a subgroup of patients achieving poor, if any, clinical benefit from everolimus-exemestane, and a second group of patients achieving long-term benefit. Specifically, a ∆SUV (%) higher than 28.77% was associated with significantly longer PFS. We also calculated the best ∆SUV (%) to discriminate LSs from non-LSs, and we found that a threshold of 53.8% was capable of identifying those patients achieving long-term benefit from the treatment.

One previously published study found that very precocious reduction (14 days after treatment initiation) of FDG uptake at PET/CT scan may predict tumor response to everolimus plus exemestane more reliably than standard CT imaging [22]. Different from this study, in our current research, we looked for the best ∆SUV (%) that is not only capable of discriminating patients benefiting from everolimus-exemestane but, more specifically, those patients achieving long-term benefit from this therapy. For the definition of LRs, we used the median PFS achieved in the experimental arm of the BOLERO-2 trial, which demonstrated the superiority of everolimus-exemestane combination over exemestane alone in terms of PFS [23]. In this study, we considered this PFS duration as sufficiently long to define a subgroup of patients.

If confirmed by future larger studies, these results could be helpful in precociously identifying patients significantly more likely to achieve long-term benefit with everolimus-exemestane.

In other solid tumors, FDG/PET have a central position in assessing response to a specific treatment [24]. In NSCLC many reports indicated that early FDG-PET/CT can predict patient response to TKI [25].

The meta-analysis performed by Jun Ma et al. showed that early metabolic response was statistically associated with improved OS (HR = 0.54) in patients with advanced NSCLC treated with TKIs [26]. Tomonobu et al. investigated the relationship between the reduced uptake in early FDG-PET/CT examination and clinical outcomes in patients with NSLCL treated with gefitinib or crizotinib [27]. Despite the small number of cases, their results indicated that the early reduction in the FDG uptake in the tumor mass after initiation of molecular agents could predict the subsequent tumor response in patients with *EGFR*-mutated or *ALK*-rearranged NSCLC [27]. Furthermore, FDG-PET/CT could serve as a prognostic tool in patients with advanced medullary thyroid cancer scheduled to undergo vandenatinib treatment. An elevated glucose consumption, as measured by FDG-PET/CT at baseline, was associated with shorter PFS, therefore indicating that these patients need to be monitored more closely in comparison to the low ^18^FDG uptake at baseline counterpart [28].

The primary novelty of our study consists in the fact that ∆SUV (%) is a simple, widely available biomarker derived from FDG/PET scans, which are commonly used for the evaluation of patient response to treatments. Additionally, we found that basal SUV was not an indicator for later response and longer survival (AUC values were not informative). These results showed the importance of longitudinally collecting SUV values (not only basal ones) in order to examine the dynamic changes and to investigate whether changes in SUV value would confer a difference in the survival outcomes, as this study had revealed this is case.

Even the data, obtained in two independent cohorts, matched, our report has several limitations, firstly the small sample size and secondly its retrospective nature. However, this is a hypothesis-generating study, therefore these results could require additional validation. Furthermore, the value of FDG-PET as an early read-out might be limited by the fact that very few patients are classified as PD after three months. As such, it was not possible to identify patients with 100% chance of early progression.

## 4. Materials and Methods

### 4.1. Patients and Study Design

The present study was a retrospective study (Protocol N. 41499) that enrolled HR+ HER2− mBC patients treated with everolimus and exemestane in two Italian centers (ASST of Cremona and Fondazione IRCCS Istituto Nazionale dei Tumori di Milano) between May 2013 and March 2018. Patients received oral everolimus (at a dose of 10 mg), plus exemestane (at a dose of 25 mg) with continuous daily dosing. The treatment was continued until disease progression, an unacceptable level of toxic effects, withdrawal of consent, loss to follow-up, or death. Dose reduction of everolimus was permitted according to the schedule of changes. Patients with the following characteristics were included: histologically confirmed HR+HER2− mBC; treatment with everolimus plus exemestane as a first or subsequent line of treatment; availability of FDG PET scans obtained at baseline and after 3 months of everolimus-exemestane treatment. Everolimus is metabolized in the liver. In particular, it is a substrate for CYP3A4, CYP3A5, CYP2C8 and P-glycoprotein [29]. For these reasons, patients receiving concomitant medications recognized as being strong inhibitors or inducers of CYP3A isoenzyme were excluded. Other exclusion criteria were: uncontrolled diabetes mellitus, pregnancy, age younger than 18 years old and previous treatment with any PI3K, AKT, or mTOR (mechanistic target of rapamycin) inhibitor. Furthermore, all patients had examined the blood glucose concentrations under the recommended guideline, in order to reduce SUVmax variability [30]. The study protocol was approved by Comitato Etico Val Padana—Cremona-Lodi-Mantova(code 12063/2015, 30/04/2015). All patients provided written informed consent for use of demographic data and medical history data. The study was performed in accordance with the Declaration of Helsinki.

### 4.2. FDG-PET/CT Acquisition and Analysis

^18^FDG-PET/CT scans were carried out using a Discovery ST PET/CT scanner (GE Healthcare, Milwaukee, WI, USA), which combines a four or eight multi-slice helical CT scanner with a PET tomography consisting of 10,080 BGO crystals arranged in 24 rings. The crystal dimensions are 6.3 × 6.3 × 30 mm^3^ and they are organized in blocks of 6 × 6 crystals, coupled to a single photomultiplier tube with four anodes. The 24 rings of the PET system allow 47 images to be obtained, spaced by 3.27 mm, and covering an axial field of view of 157 mm. The average transverse and axial spatial in-plane image resolution was approximately 6 mm full-width at half-maximum (FWHM) at 1 cm and 10 cm off-axis.

For this study, two PET scans were performed in all patients: baseline and 3 months after everolimus-exemestane treatment. ^18^FDG-PET/CT images were evaluated independently by two experienced radiologists. The same PET scanner, display station, and protocols for preparation, acquisition, processing and analysis were used for baseline and after 3 months treatment. Patient scans were acquired with uptake durations that are within ±15 min of each other. Standardized uptake values (SUVs) were calculated. The metabolic response was calculated using the following formula: ΔSUV% [(SUV baseline-SUV 3 months)/SUV baseline] × 100%.

### 4.3. Statistical Analyses

The primary objective was to identify the best ∆SUV% that could predict long-term clinical benefit from everolimus-exemestane treatment. Descriptive statistics were used to analyze and report patients’ characteristics. Wilcoxon test for paired data was used to compare SUV at baseline and SUV after 3 months. Patients were defined as long responders (LRs) if disease progression occurred at least 10 months after treatment initiation, whereas patients progressing before 10 months after treatment initiation were defined as non-LRs [23]. Receiver operating characteristic (ROC) curve analysis was performed to identify the best cut-offs value of ΔSUV% that were capable of differentiating LRs from non-LRs and LSs from non-LSs. Optimal cut-off values were determined using the Youden Index (J), as defined as maxc {Sensitivity (c) + Specificity (c) − 1}. The cut-point that achieves this maximum value is referred to as the optimal cut-point (c*) because it is the cut-point, which optimizes ΔSUV% differentiating ability when equal weight is given to sensitivity and specificity. Progression-free survival (PFS) was used as the primary clinical endpoint. PFS was defined as the time (months) from start of treatment until disease progression or death, whatever the cause, or last follow-up and was estimated by Kaplan–Meier method. The median follow-up was computed for censored patients, excluding women with the events of interest (reverse Kaplan–Meier method). Overall survival (OS) was defined as the time (months) from the first cycle of everolimus-exemestane treatment to the date of patient death or last contact. Patients alive were censored at the last time point. Based on the ΔSUV% (defined by cut-off values), we evaluated through the log-rank test survival outcomes in LSs and non-LSs.

## 5. Conclusions

In the era of precision oncology, finding new biomarkers of clinical benefit from current therapies is of paramount importance. Treatment personalization is especially important in the case of costly and potentially toxic targeted therapies, such as the everolimus-exemestane combination. In this clinical setting, identifying those patients more likely to achieve long-term clinical benefit from everolimus could help clinicians to identify the subgroup of HR+ HER2- mBC patients most likely to benefit from it, while using potentially more effective therapies in the remaining patients. Here, we found a threshold of ∆SUV that is capable of precociously identifying HR+ HER2- mBC patients who are much more likely to benefit from everolimus-exemestane in the long-term period.

Nevertheless, the role of ^18^FDG-PET/CT in this setting deserves to be further investigated and image-derived biomarkers require a clinical validation before entering in clinical practice. Besides that, if our data is confirmed by future larger studies, this could have an impact on treatment personalization in HR+ HER2- mBC patients.

## Figures and Tables

**Figure 1 cancers-12-03314-f001:**
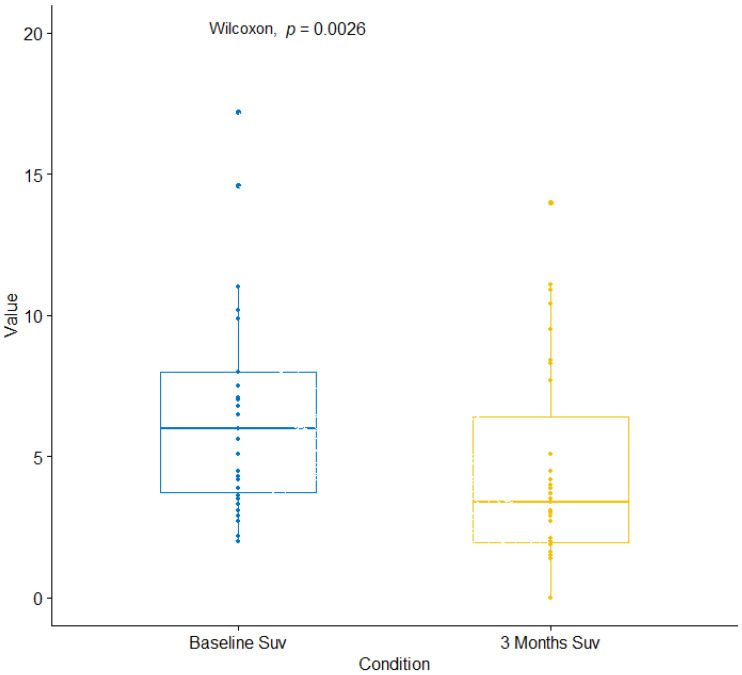
Box plot diagram of distribution of SUV max values at baseline and at 3 months.

**Figure 2 cancers-12-03314-f002:**
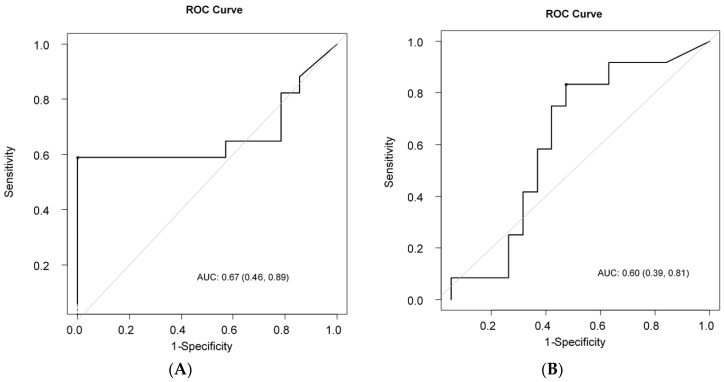
(**A**) Receiver-operating characteristic (ROC) analysis to differentiate LRs from non-LRs. Cut-off Δ (SUV) (%): 28.8%, area under the curve: 0.67; (**B**) ROC analysis to differentiate LSs from non-LSs. Cut-off Δ (SUV) (%): 53.8%, area under the curve: 0.62.

**Figure 3 cancers-12-03314-f003:**
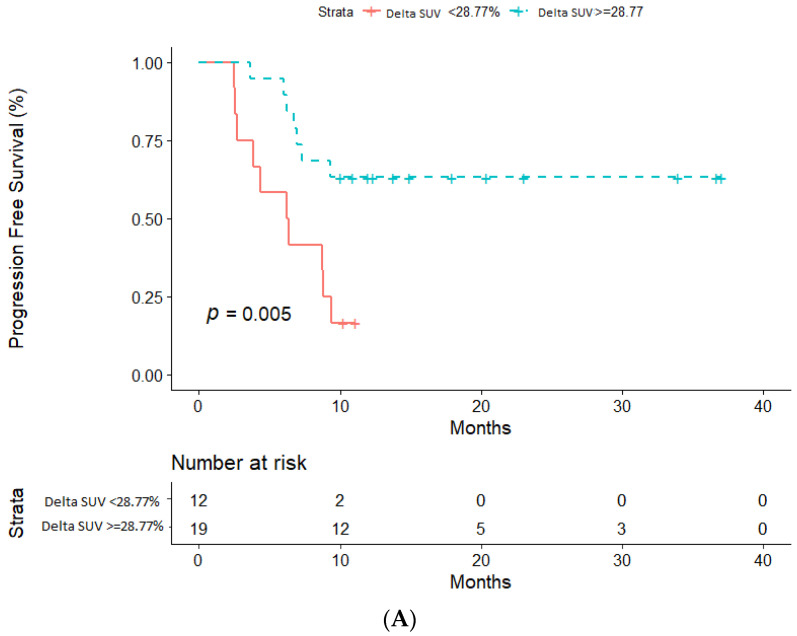
Kaplan–Meier survival estimates based on differences in metabolic response at three months. (**A**) Progression Free Survival in non-LRs (**B**) Overall survival in non-LSs.

**Figure 4 cancers-12-03314-f004:**
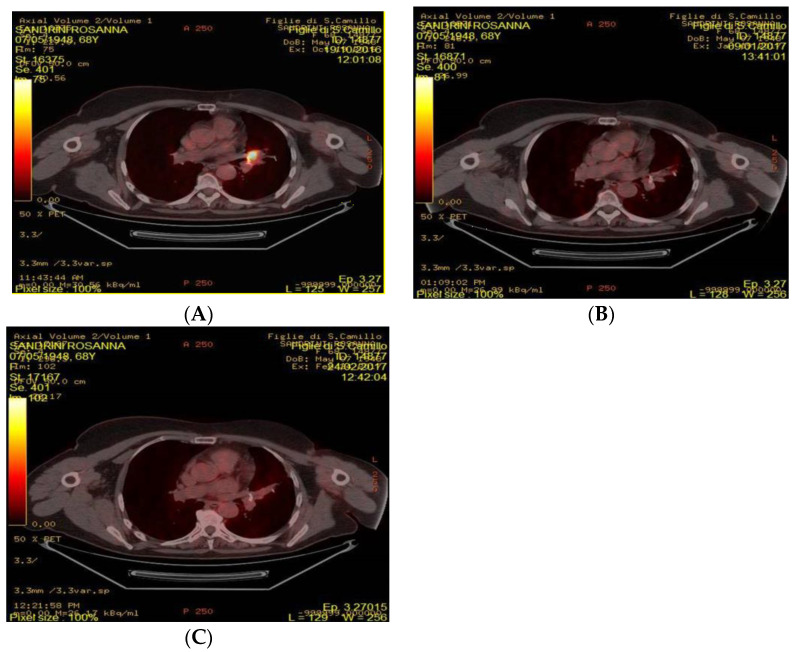
FDG-PET performed at baseline, after one month and after three months. (**A**) FDG-PET/CT performed at baseline: appearance of intense hyperaccumulus due to parenchymal thickening in the left peri-hilar area, with a segmental lingularcharactern (SUVmax 14.6) (**B**) FDG.PET performed after 1 months: Clear reduction of intense hyper-accumulation, however minimal residual collection remains (SUVmax 3.5) (**C**) FDG-PET/CT performed after 3 months unchanged the uptake previously reported for parenchymal thickening in the left peri-hilar area, with a segmental lingular character.

**Table 1 cancers-12-03314-t001:** Patient demographics and clinical characteristics of 31 patients included in the study.

Characteristic	Baseline Data
**Age (y)**	
Median (Range)	65.3 (48.6; 80.3)
**Baseline SUV**	
Median (Range)	6.0 (2.0; 17.2)
**3 months SUV**	
Median (Range)	3.4 (0.0; 14.0)
**ΔSUV% 3 months**	
Median (Range)	39.2% (−375.0%; 100%)
**Duration of Everolimus Treatment**	
<10 months	22 (71.0%)
≥10 months	9 (29.0%)
**Follow-up (months)**	
Median (Range)	17.92 (12.96–21.69)

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
