# Peer review of "Early Changes of the Standardized Uptake Values (SUVmax) Predict the Efficacy of Everolimus-Exemestane in Patients with Hormone Receptor-Positive Metastatic Breast Cancer"

_cancers, 2020, doi:10.3390/cancers12113314_

Round 1
Reviewer 1 Report
Sirico et al studied FDG-PET uptake as a marker of long term response to everolimus-exemestane therapy in patients with HR+ metastatic breast cancer.
They reported treshold for FDG-PET deltaSUV as a way to identify patients with long response and long survival outcomes. Following are my questions:
1-Can the authors comment on the PIK3CA mutation state of the patients that might not respond to the therapy?
2-Is there any information on the initial PI3K or mTOR signaling state of these tumors?
3-In certain cases the tumors that don't uptake FDG-PET might switch to glutamine signaling. Is there anyway to normalize uptake signal to the tumor size? It might improve current AUC values, which are quite low.
4-Are the initial/basal FDG-PET uptake values (basal SUV) indicator for later response or sustaining responsiveness in this data set? What would be their AUC ROC compared to that of deltaSUV?
Author Response
1-Can the authors comment on the PIK3CA mutation state of the patients that might not respond to the therapy?
RE: We thank the reviewer for this comment. Unfortunately we did not conduct a pathologic assessments of PIK3CA mutation state. At that time we designed the study, there was only few evidences showing correlation between PIK3CA mutations and response to everolimus [1]. Our study was conducted after the results of the BOLERO-2 trial; Hortobagyi et al performed a next-generation sequencing (NGS) analysis in archivial tumor specimens from a subgroup of patients enrolled in the BOLERO-2 trial showed that Everolimus provides clinical benefit to patients with both PIK3CA-wild type and PI3KCA mutated tumors [2]. Therefore, our choice was based on the lack of prospective data, which could not allow for a clean interpretation of whether drug sensitivity is mediated by the presence or absence of a PI3KCA mutation or set of mutations. In the next upcoming future, thanks to the data showed in the SOLAR-1 study, it would be important to test PI3KCa status as a surrogate markers to identify patients would have be benefit from alpelisib (an PI3KCA inhibitor) or everolimus (mTOR inhibitor) in clinical practice [3, 4].
2-Is there any information on the initial PI3K or mTOR signaling state of these tumors?
RE: Thank for the question.
From the patients enrolled in the study, we were able to collect 16 basal tumor FFPE. We found that 5 of them were PI3KCA mutated. Specifically one mutation (C420R) is located in exon, one in exon 20 (H1049C), one in exon 4 (N345K) and two identical mutations in exon 9 (E545K). The other basal tumor FFP were PI3KCA wild type. On the other hand, we managed to collect only 5 metastasis tumor FFPE (in these cases tumor biopsy was not practicable), finding that only two sample had PI3KCA mutated: one in exon 1 (R93W) and the other one in exon 4 (N345K). However, the two tumor sample mutated, derived both from PI3KCA wild type tumor basal sample. Therefore, PI3KCA mutation status changes from baseline tumor to metastasis. In addition, 16 patients underwent ctDNA analysis before treatment and none PIK3CA gene mutations were detected. Unfortunately, the numbers of patients evaluated is too small to perform any adequate statistical analysis.
3-In certain cases the tumors that don't uptake FDG-PET might switch to glutamine signaling. Is there anyway to normalize uptake signal to the tumor size? It might improve current AUC values, which are quite low.
RE: The question posed is very interesting and stimulating. However, we do not have the tumor size with the regard of the target lesion from which we have the SUV related data. The CT scan used for the PET-CT scan was imprecise in evaluating the tumor size and it was too much operator dependent; whereas the SUV detection was monitored by a specific software present in both sites with a higher chance to compare the obtained data. In one site, the PET-CT scan report showed SUV only and not tumor dimension
with consequent difficulties to re-test them. We will take the suggestion into account for our ongoing studies on the role of PET-CT scan to monitor tumor response.
4-Are the initial/basal FDG-PET uptake values (basal SUV) indicator for later response or sustaining responsiveness in this data set? What would be their AUC ROC compared to that of deltaSUV?
We thank the reviewer for this question: we have performed these requested analyses and we found that in our study, basal SUV is not an indicator of later response nor longer survival.
In particular, AUC values for basal SUV indicated that basal SUV has a poor ability to discriminate LRs vs nLRs (AUC=0.50 (0.29 - 0.72)). This AUC value is lower than the one computed with Delta SUV (AUC=0.67), even if the difference is not statistically significant (p=0.28, De Long Test)
Similarly AUC values for basal SUV indicate that basal SUV has a poor ability to discriminate LSs vs nLSs (AUC=0.48 (0.26 - 0.69)). This AUC value is lower than the computed with Delta SUV (AUC=0.60), even if the difference is not statistically significant (p=0.42, De Long Test)
These results confirmed the aim of our study because they showed that it is important longitudinally to investigate SUV values (not only basal ones) in order to examine the dynamic changes and to investigate whether changes in SUV value would confer a difference in the survival outcomes.
We integrated the revised paper in the sections (from line 152 to 154).
Does the introduction provide sufficient background and include all relevant references?
Can be improved
Regarding this question we have implemented:
-Introduction: from line 77 to 90
from line 109 to 120
We have also implemented:
- Discussion: from 231 to 236
-Results: from line 146 to 148
|
Are the conclusions supported by the results? |
We implemented this part from line 311 to 313
Submission Date
23 September 2020
Date of this review
23 Oct 2020 16:49:28
References
- Mohseni M, Park BH. PIK3CA and KRAS mutations predict for response to everolimus therapy: Now that’s RAD001. J. Clin. Invest. 2010. doi:10.1172/JCI44026.
- Hortobagyi GN, Chen D, Piccart M et al. Correlative analysis of genetic alterations and everolimus benefit in hormone receptor-positive, human epidermal growth factor receptor 2-negative advanced breast cancer: Results from BOLERO-2. J. Clin. Oncol. 2016. doi:10.1200/JCO.2014.60.1971.
- Schettini F, Buono G, Trivedi M V. et al. PI3K/mTOR Inhibitors in the Treatment of Luminal Breast Cancer. Why, When and to Whom? Breast Care 2017. doi:10.1159/000481657.
- André F, Ciruelos E, Rubovszky G et al. Alpelisib for PIK3CA-mutated, hormone receptor-positive advanced breast cancer. N. Engl. J. Med. 2019. doi:10.1056/NEJMoa1813904.
Reviewer 2 Report
SUV max predict the efficacy of everolimus-exemestane in ER+ BC is neither a predictive or prognostic marker.
There are several molecular biomarkers are there and everolimus is using routinely in the clinic. On top of that everolimus is not a very toxic agent too.
Major Criticism:
- Do not understand SUVmax is predictive or prognostic biomarker?
- What is the average SUV and is there any relationship with Metabolic Target Volume especially in the metastatic sites e.g. lung or liver
- Patients numbers are less
- Line 40-46 is not clear to the reader. Needs clear clarification
- FDG-PET (delta SUV%): Is it correlate with the upregulation of the PI3K-AKT-mTOR pathway?
- There is huge number of typo errors. It expresses sever negligence.
- Figure quality is not good
- Methodology is extremely brief.
Author Response
- Do not understand SUVmax is predictive or prognostic biomarker?
We thanks the reviewer for this clarification asked.
In the text we have incorrectly used the term predictive and in the new version of the paper we have
removed this term.
In our study we analyze only the group of patients who all received the treatment of interest (everolimus-exemestane) and demonstrate that biomarker-positive (∆SUV(%) higher than 28.77 %) patients have better outcomes compared with biomarker-negative ( ∆SUV(%) lower than 28.77 %) patients. In this case, there is no comparison group and so a formal statistical test for interaction between the treatment and biomarker cannot be performed.
For this reason we are not able to assess that ∆SUV is predictive nor prognostic marker, but we have showed that that it could be useful to stratify the patients in two group (LRs and non LRs) based on differences in metabolic response, guiding toward precision medicine discovery.
- What is the average SUV and is there any relationship with Metabolic Target Volume especially in the metastatic sites e.g. lung or liver?
We did not collect this data because at our Hospital, we do not used MTV routinely. Although intensive research on MTV showing promising results, MTV is not used in standard clinical practice yet, probably because it is requires an accurate segmentation of the tumor [1], unlike SUVmax, and currently there is no consensus on the optimal method to segment tumors in FDG PET image [2]. Moreover Kornélia Kajárya et al showed that SUVmax may reflect tumour metabolism more reliably compared with MTV [3]. Probably further large-scale prospective studies are needed in order to confirm the value of these parameter. On the other hand, SUVmax is a simple, observer-independent and highly repeatable imaging biomarker that is ideally suited for monitoring tumor response to treatment in patients.
- Patients numbers are less
As reported in the paper, the small sample size is a limitation of our study. However, to support our study, it is important to state all patients included in the study had a very well defined biological/histologically characteristic of the tumor and clinical setting: hormone receptor-positive (HR+) human epidermal growth factor receptor 2-negative (HER2-) metastatic breast cancer (HR+ mBC) progressing on prior therapy with a non-steroideal aromatase inhibitor.
- Line 40-46 is not clear to the reader. Needs clear clarification
We thank the reviewer for this observation:
We have modified the abstract from line 40 to 46, so that it is now easier to read
- FDG-PET (delta SUV%): Is it correlate with the upregulation of the PI3K-AKT-mTOR pathway?
In our paper we could not evaluated a correlation between ΔSUV and an eventually upregulation of the PI3K-AKT-mTOR pathway, because we did not conduct, before starting the treatment, a pathological assessment of PI3K status on tumor tissue. At the time, the data of the role of PI3Kca status and response to everolimus were controversial [4].
Recently Heinrich Magometschnigg et al investigate the functional association of PIK3CA mutational status and tumor glycolysis assessed by FDG-PET(using maximum standardized uptake values (SUVmax)). In this prospective study they found that patients who harbored a PIK3CA mutation had a higher median [18F]FDG-PET, compared to patients with WT PIK3CA; however, this difference was not significant (p = 0.07). Beside, from what we know from literature, glucose uptake is independently associated with PIK3CA mutations [5]. Regarding the other mutations involved in PI3K/AKT/mTORC1 axis, as AKT, mTORC1 and PTEN, there is no data in literature that show a correlation between mutations in these gene and FDG-PET. We have tested the PI3KCa status, retrospectively; unfortunately, the numbers are too small to drive any adequate statistical conclusions and any speculations.
- There is huge number of typo errors. It expresses sever negligence.
The revised version of the manuscript has been read and improved as regard as quality of English and typo errors
- Figure quality is not good:
We have redone the graphics improving the quality and resolution
Line: 134-135
Line: 149-150
Line: 151-152
Line: 168-169
Line: 170-171
- Methodology is extremely brief.
We have implemented methodology:
- Patients and Study design: Line: from 247 to 250 and from 256 to 259
- FDG-PET/CT Acquisition and Analysis: Line: from 265 to 277
|
Does the introduction provide sufficient background and include all relevant references? Must be improved:
- Discussion: from 231 to 236 - Results: from line 146 to 148 |
( ) |
|
Are the conclusions supported by the results? We implemented this part from line 311 to 313 |
( ) |
(x) |
References
- Moon SH, Hyun SH, Choi JY. Prognostic significance of volume-based PET parameters in cancer patients. Korean J. Radiol. 2013. doi:10.3348/kjr.2013.14.1.1.
- Im HJ, Bradshaw T, Solaiyappan M, Cho SY. Current Methods to Define Metabolic Tumor Volume in Positron Emission Tomography: Which One is Better? Nucl. Med. Mol. Imaging (2010). 2018. doi:10.1007/s13139-017-0493-6.
- Kajáry K, Tokés T, Dank M et al. Correlation of the value of 18F-FDG uptake, described by SUVmax, SUVavg, metabolic tumour volume and total lesion glycolysis, to clinicopathological prognostic factors and biological subtypes in breast cancer. Nucl. Med. Commun. 2015. doi:10.1097/MNM.0000000000000217.
- Hortobagyi GN, Chen D, Piccart M et al. Correlative analysis of genetic alterations and everolimus benefit in hormone receptor-positive, human epidermal growth factor receptor 2-negative advanced breast cancer: Results from BOLERO-2. J. Clin. Oncol. 2016. doi:10.1200/JCO.2014.60.1971.
- Magometschnigg H, Pinker K, Helbich T et al. PIK3CA Mutational Status Is Associated with High Glycolytic Activity in ER+/HER2− Early Invasive Breast Cancer: a Molecular Imaging Study Using [18F]FDG PET/CT. Mol. Imaging Biol. 2019. doi:10.1007/s11307-018-01308-z.
Round 2
Reviewer 1 Report
Thanks for addressing the questions we posed.
Author Response
Thank you. We have double checked the spelling as we would do per proof reading.
Reviewer 2 Report
I have noticed a substantial improvement in this revised version. Now it is well shaped/publishable
article.
Following minor correction is needed
- Please rectify the punctuation/ spacing error
- Authors mentioned long term survival, is it mean disease free survival (DFS)? Please clear it.
- Line 79-88, totally known fact. Please make it shorter.
- In line 89-90, all AKT inhibitors (e.g. AZD5363 or GDC-0068) are in late phase of clinical trial. Please change the sentences and put appropriate references.
- Please 18 FDG , make it similar to everywhere (not 18FDG and 18 FDG.
- In line 121, authors mentioned "assumptions", please change it to "reports".
- In line 229 instead of "main novelty", please change it to "primary novelty".
- In line 258, patients had blond glucose concentration, please add patients had examined the blood glucose concentration
Author Response
Thank you. Please find point-by-point responses:
Following minor correction is needed
- Please rectify the punctuation/ spacing error
We corrected punctuation and spaces.
- Authors mentioned long term survival, is it mean disease free survival (DFS)? Please clear it.
We explained in simple summary and main text the term longer survival as in the abstract (36 month-OS).
- Line 79-88, totally known fact. Please make it shorter.
I made it shorter.
- In line 89-90, all AKT inhibitors (e.g. AZD5363 or GDC-0068) are in late phase of clinical trial. Please change the sentences and put appropriate references.
We fixed it: Currently, PI3K and mTOR inhibitors as well as inhibitors of AKT (e.g. AZD5363 or GDC-0068) are in late phase clinical trials [13] (NCT01485861, NCT03012477).
- Please 18 FDG , make it similar to everywhere (not 18FDG and 18 FDG.
We fixed it.
- In line 121, authors mentioned "assumptions", please change it to "reports".
We fixed it.
- In line 229 instead of "main novelty", please change it to "primary novelty".
We fixed it.
- In line 258, patients had blond glucose concentration, please add patients had examined the blood glucose concentration
We fixed it.